# Ultrasound and Microbubbles Mediated Bleomycin Delivery in Feline Oral Squamous Cell Carcinoma—An In Vivo Veterinary Study

**DOI:** 10.3390/pharmaceutics15041166

**Published:** 2023-04-06

**Authors:** Josanne S. de Maar, Maurice M. J. M. Zandvliet, Stefanie Veraa, Mauricio Tobón Restrepo, Chrit T. W. Moonen, Roel Deckers

**Affiliations:** 1Imaging and Oncology Division, University Medical Center Utrecht, Utrecht University, 3508 GA Utrecht, The Netherlands; 2Department of Clinical Sciences, Faculty of Veterinary Medicine, Utrecht University, 3584 CL Utrecht, The Netherlands

**Keywords:** USMB, bleomycin, contrast-enhanced ultrasound, veterinary medicine, companion animals, feline, oral squamous cell carcinoma, head and neck cancer

## Abstract

To investigate the feasibility and tolerability of ultrasound and microbubbles (USMB)-enhanced chemotherapy delivery for head and neck cancer, we performed a veterinary trial in feline companion animals with oral squamous cell carcinomas. Six cats were treated with a combination of bleomycin and USMB therapy three times, using the Pulse Wave Doppler mode on a clinical ultrasound system and EMA/FDA approved microbubbles. They were evaluated for adverse events, quality of life, tumour response and survival. Furthermore, tumour perfusion was monitored before and after USMB therapy using contrast-enhanced ultrasound (CEUS). USMB treatments were feasible and well tolerated. Among 5 cats treated with optimized US settings, 3 had stable disease at first, but showed disease progression 5 or 11 weeks after first treatment. One cat had progressive disease one week after the first treatment session, maintaining a stable disease thereafter. Eventually, all cats except one showed progressive disease, but each survived longer than the median overall survival time of 44 days reported in literature. CEUS performed immediately before and after USMB therapy suggested an increase in tumour perfusion based on an increase in median area under the curve (AUC) in 6 out of 12 evaluated treatment sessions. In this small hypothesis-generating study, USMB plus chemotherapy was feasible and well-tolerated in a feline companion animal model and showed potential for enhancing tumour perfusion in order to increase drug delivery. This could be a forward step toward clinical translation of USMB therapy to human patients with a clinical need for locally enhanced treatment.

## 1. Introduction

Head and neck squamous cell carcinoma (HNSCC) is diagnosed in approximately 900,000 patients annually worldwide, 5% of all cancer diagnoses [1]. Circa 450,000 patients die of HNSCC yearly. Most patients present with locally advanced disease [2,3], and because primary surgery is often not possible or is expected to result in unacceptable morbidity, they are often treated with combination therapies, including radiotherapy, chemotherapy and targeted therapy [4]. Even so, up to half of the patients develop (often incurable) local recurrences [5,6] and treatment is associated with acute and long-term toxicity [7,8,9]. Primary chemoradiotherapy with cisplatin as a radiosensitizer is often used in locally advanced HNSCC. A higher cumulative cisplatin dose is associated with better local control and, to some extent, longer overall survival [7,10], but due to local and systemic toxicities, 30–50% of patients cannot complete all planned cycles of cisplatin [7,11]. This emphasizes the need for improved local tumour delivery of cisplatin, without increasing the chemotherapy dose in healthy tissues. Consequently, a method to increase local drug delivery without increasing systemic toxicity could lead to improved outcomes of (chemo)radiotherapy.

We hypothesize that this goal could be achieved by means of ultrasound and microbubbles (USMB) therapy. Microbubbles are micron-sized gas-filled bubbles used for contrast-enhanced ultrasound (CEUS) imaging [12,13]. When exposed to ultrasound, stable or inertial cavitation of microbubbles can occur, creating a number of biological effects collectively termed sonopermeation [14,15]. USMB therapy has been shown to improve drug delivery for various molecules in vitro and in vivo [15]. In particular, addition of USMB therapy improved the effect of chemotherapy or chemoradiotherapy with cisplatin on HNSCC cells in vitro [16]. However, pre-clinical studies often use custom-made set-ups and a large variety of US parameters, which cannot easily be translated to the clinic. Most (ongoing) clinical studies of USMB therapy have focused on brain applications, using dedicated systems not applicable to other organs [17,18,19]. Meanwhile, studies outside the brain are limited to small numbers of patients with pancreatic cancer, liver metastases and breast cancer, which all used different ultrasound settings [20,21,22,23]. Thus far, USMB therapy has not been studied in human head and neck cancer patients. Using clinically available US systems and settings (preferably uniform US settings across clinical studies) combined with FDA/EMA-approved microbubbles can help make this technique more accessible.

In order to bridge the gap between human HNSCC patients and in vitro and laboratory (rodent) animal studies, we performed an in vivo veterinary feasibility trial in non-laboratory veterinary patients. Cats are a suitable model because they are large enough to use clinical ultrasound equipment, and because many pathophysiological and genetic similarities exist between humans and cats [24,25,26]. Furthermore, feline oral squamous cell carcinoma (FOSCC) is very common in aged cats [27]. Like human patients, cats often present with advanced stage disease (e.g., with muscle or bone invasion) [28] and often succumb to local disease progression, rather than metastatic disease [26,29]. Standard-of-care treatment options are similar to the human setting: when surgery is not an option, primary radiotherapy can be combined with chemotherapy as a radiosensitizer [30,31,32]. Supportive care (antibiotics and/or anti-inflammatory drugs) results in a median overall survival of approximately 44 days [33]. The platinum-based drug cisplatin, often used in human patients, creates unacceptable toxicity in cats when systemically administered [34]. Instead, bleomycin is a well-tolerated cytostatic drug in cats, but has limited efficacy because efficacy is dependent on intracellular uptake, which is complicated by its hydrophilic nature and dependence on protein receptors to enter the cell [35,36,37]. Bleomycin is very toxic inside the cell, by inducing oxidative damage leading to DNA single- and double-strand breaks. It is partly inactivated by the enzyme bleomycin hydrolase, which is present in most tissues (but less abundant in skin and lungs, locations of bleomycin toxicity) and excreted primarily via the kidneys (mean plasma drug clearance in humans is 70 mL/min/m^2^ and strongly correlated with renal function) [36,37,38]. To improve the efficacy of bleomycin, electroporation can be used, also known as electropermeabilization: making cell membranes reversibly permeable by application of an electrical current [39,40]. In a feline study the combination of bleomycin plus electroporation resulted in an overall response of 89%, compared to 33% with bleomycin alone [41], but it causes unpleasant muscle contractions and possibly an increased risk of cardiac arrhythmias [41,42]. After electroporation of large bulky tumours, patients could be more susceptible to adverse events such as tumour lysis syndrome, thromboembolism, disseminated intravascular coagulation, delayed wound healing and local necrosis [40]. These adverse effects have not been described for USMB therapy. A safety study performing USMB therapy on the livers of eight pigs with a clinical US system did not result in any clinical adverse events or histopathological damage to the liver [43]. Meanwhile, CEUS imaging has been studied in hundreds of cats, without significant adverse effects [44,45,46,47,48,49]. In vitro, USMB therapy was shown to enhance local bleomycin effects [50,51,52], also when using a clinical US system with standard settings and clinically available microbubbles [53].

The combination of bleomycin with USMB therapy using a clinically available US system and microbubbles could provide a low-toxicity, low-burden additional treatment option for these cats, as well as a step towards clinical translation to human head and neck cancer patients. The primary objectives of this veterinary study were to evaluate tolerability and feasibility of bleomycin plus USMB therapy while using a clinical US system and microbubbles, while secondary objectives were to assess tumour response, survival and the effect of USMB therapy on tumour perfusion.

## 2. Methods

### 2.1. Subjects

Six cats with spontaneously arisen FOSCC were eligible for inclusion in our single-arm prospective study. They had at least cytologically proven squamous cell carcinoma, without other suitable treatment options except for palliative care, and informed consent was provided by the pet owner. Exclusion criteria were life-threatening comorbidities leading to a life expectancy of less than 1 month, contraindications for anaesthesia and known hypersensitivity to bleomycin or any of the excipients of SonoVue (Bracco, Geneva, Switzerland).

### 2.2. USMB Treatment

Each cat was treated three times, once per week (see Figure 1 for timeline of treatment procedures). US imaging and treatment were performed using an EPIQ5 or EPIQ7 imager with a C9-2 transducer (Philips, Best, The Netherlands) complemented by an L18-5 transducer solely for US imaging. A tissue-mimicking gel was used to obtain enough distance between probe and cat for the region of interest to be outside the near field of the transducer. The treatment was performed under general anaesthesia, while continuously monitoring vital signs. USMB therapy was started 7 min after intravenous (i.v.) injection of bleomycin (10,000 IU/m^2^). Microbubbles (SonoVue, Bracco, conc. 1–5 × 10^8^ bubbles/mL, dosage 0.1 mL/kg body weight per bolus injection) followed by a 1.5 mL saline flush were administered through an i.v. catheter of at least 22 gauge. When the microbubbles appeared in the tumour (based on CEUS imaging), treatment of the oral tumour was started in Pulse Wave (PW) Doppler mode. PW Doppler for 15 s was alternated with CEUS imaging for 5 s (to allow for complete reperfusion of the tumour with fresh microbubbles) and repeated five times per MB injection, as by that time no more MBs were visible on CEUS images. This process was repeated three times to a total of four microbubble injections for therapy. During USMB therapy the probe was hand-held by a veterinary radiologist and the Sample Volume of the PW Doppler was slowly moved to treat the entire tumour. Before a new microbubble injection, transducer orientation was changed to treat a different cross-section of the tumour. Optimized PW Doppler settings for USMB therapy and CEUS parameters used in cats 2–6 are shown in Table 1. These procedures were optimized during treatment of cat 1 and the first treatment session of cat 2.

### 2.3. Other Study Procedures

Primary endpoints were tolerability and feasibility, assessed at baseline and 1, 2 and 5 weeks after the first USMB therapy. Tolerability was assessed by reporting adverse events, clinical performance score and quality of life. Adverse events were reported by VCOG Common Terminology Criteria for Adverse Events version 1.1, clinical performance (CPS, 0–5 [54]) was monitored and quality of life (QoL) was assessed using a 16-item owner-completed questionnaire, translated to Dutch with permission from Adelphi UK and Zoetis [55]. Feasibility was assessed by the amount of time needed for study procedures and the ability to complete study treatments per cat. Secondary endpoints were clinical response (including tumour response and survival) and the effect of USMB therapy on tumour perfusion. Tumour response was evaluated by calliper measurements at baseline and 1, 2 and 5 weeks after the first USMB therapy. Overall survival was registered from first USMB therapy until death. To evaluate the effect of USMB on tumour perfusion, CEUS was performed immediately before and after USMB treatment using a mechanical arm to position the US probe and the same volume of MBs used for USMB therapy. All equipment settings (MI = 0.06, gain = 45%, TGC at central position, dynamic range = 50, persistence set to Off, see Table 1 for more details) were kept consistent for all cats and between CEUS measurements before and after USMB therapy.

### 2.4. Quantitative CEUS Evaluation

Time—intensity curve analysis was performed on contrast loops before and after USMB treatment to evaluate changes in tumour perfusion using in-house developed Matlab software. Image data post-processing consisted of 5 steps. First, the DICOM images, transferred from the ultrasound imager, were loaded using the standard Matlab DICOM reader and the colour images were converted into greyscale. Second, the onset of the contrast enhancement was determined. Third, from the images obtained before the onset of contrast enhancement, an averaged image (background image) was calculated and this background image was subtracted from the original images. Fourth, the temporal data were smoothed using a moving-average filter. Finally, peak intensity (PI: maximum signal intensity, also known as peak enhancement), time to peak (TTP: time between first arrival of contrast and reaching maximum intensity) and area under the curve (AUC: area under the time versus signal intensity curve) maps were calculated for each pixel [56]. Using a region of interest (ROI) with the same size before and after USMB therapy, the parameters were visualized with a colour scale and plotted in histograms to compare results pre and post USMB therapy. Pixels which already had high signal intensity before onset of the contrast enhancement (e.g., regions containing bone) were excluded for analysis. The percentage of pixels reaching >25% of the peak intensity in that ROI were compared between before and after USMB with a paired samples Wilcoxon test. A two-sided *p*-value < 0.05 was considered significant. Median values of the AUC in the ROI were compared after discarding AUC values ≤ 0.

## 3. Results

### 3.1. Baseline Characteristics

We included six feline patients with tumours located in different oral regions.

Table 2 describes relevant patient and tumour characteristics. All cats were domestic shorthairs over 10 years of age, with T2 or T3 tumours [57], two cats had known lymph node metastases and one cat had pulmonary metastases. Only cat 6 had received previous surgical treatment and was included after finding that the tumour had not been removed entirely. All cats received supportive care with antibiotics and pain medication, and two cats required assisted feeding (oesophageal feeding tube). Three cats received concomitant treatment for hyperthyroidism (thiamazole or carbimazole).

### 3.2. Bleomycin plus USMB Therapy Was Tolerable

The USMB treatments were well tolerated. All cats experienced adverse events, all except one were not severe and were primarily grade 1 or 2 (Table 3 lists all adverse events). In addition, adverse events were considered related to anaesthesia (e.g., constipation, fatigue, hypotension, hypothermia, lethargy and vomiting), comorbidity (untreated hyperthyroidism) or progressive tumour growth (e.g., anorexia, generalized weakness, fatigue, haemorrhage from the tumour, pain, ptyalism, skin ulceration, soft tissue necrosis and weight loss). One cat had localized alopecia and erythema related to an i.v. catheter. Another cat experienced a mild sinus tachycardia during the treatment session, which could be related to USMB therapy or anaesthesia but resolved spontaneously. Cat 6 had small ulcers of the tongue which responded to antibiotics and later developed a functional impairment of tongue movement with dysphagia grade 2 during follow-up, which progressed to grade 3, a serious adverse event leading to euthanasia after 147 days. We saw a fibrosis-like clinical picture with chronical inflammation and epithelial proliferation in the biopsy. Cat 6 also developed soft tissue necrosis grade 1 of the tip of the tongue, without signs of tumour progression.

Clinical performance score (CPS) did not change from 0 (“fully active”) in cats 1, 2, 3 and 6. In cat 4 it temporarily decreased one point in week 1 but then recovered to “fully active”, and in cat 5 it decreased from 1 (“slight tiredness/dyspnoea after severe exertion”) to 3 (“spontaneous tiredness or dyspnoea without exertion, lies often on the floor”) after 5 weeks. Figure 2 shows QoL scores. QoL remained stable in cats 1–3, and decreased gradually in cat 5, most likely due to tumour progression. In cat 6 it decreased after one week and then stabilized. In cat 4, QoL decreased temporarily after one week, due to pain and inability to eat. Supportive treatment was intensified (antibiotics restarted, pain medication increased) and the next USMB treatment was postponed for 1 week. In this week the cat’s condition and QoL improved. Since QoL recovered quickly, infection at the tumour site was considered the most likely cause.

### 3.3. Three Treatment Sessions of Bleomycin plus USMB Therapy Were Feasible

The study treatment was considered feasible, as all six pet owners completed all planned study visits, including three USMB treatment sessions and a follow-up visit. In the 5 cats with optimized procedures, mean time in hospital per treatment session ranged from 153 min in cat 4 to 207 min in cat 3, and mean time spent on US imaging plus therapy ranged from 41 min in cat 6 to 66 min in cat 3 (Figure 3).

### 3.4. Modest Clinical Response

For cat 1, USMB therapy procedures had not yet been optimized; therefore, its clinical response parameters are not reported here. Upon clinical examination, 4 cats had stable disease during the 3 treatment sessions, but disease progression was observed at 5 weeks (cats 3 and 5) or 11 weeks after the first treatment session (cat 2). Cat 4 had progressive disease one week after the first treatment session (possibly due to tumour infection) but remained stable at follow-up. Eventually, all cats except for cat 6 had progressive disease (Figure 4). Four cats were euthanized 85, 64, 56 and 147 days after their first treatment session; the fifth cat died spontaneously after 57 days (Table 2).

### 3.5. Indication of Increased Tumour Perfusion Assessed by Contrast-Enhanced Ultrasound

CEUS imaging before and after USMB treatment was available for cat 2 (two treatment sessions), cat 3 (three treatment sessions) cat 4 (one treatment session), cat 5 (three treatment sessions) and cat 6 (three treatment sessions). During the first treatment session of cat 2, the US probe was unintentionally moved during CEUS acquisition, and during the second and third treatment sessions of cat 4, the US system with contrast license was unavailable. Figure 5 shows representative US parametric maps in cat 3 and Appendix A show an overview of all cats. Based on visual interpretation of the data, peak intensity (PI) increased in seven out of twelve treatment sessions, decreased in four and did not change in one (Appendix A), while for time to peak (TTP) the changes were small and there was no clear trend (Appendix A). The median AUC in the ROI decreased in six out of twelve treatment sessions (median decrease 39.1%, range −4 to −65%) while it increased in six out of twelve treatment sessions (median increase 199%, range 10 to 1039%) (Appendix A). When comparing the percentage of pixels reaching >25% of the peak Intensity (PI) after USMB therapy with the baseline in that treatment session, we observed an increase in 8 sessions (green line), a decrease in 3 sessions (red line) and no change in 1 one session (black line, Figure 6). These findings also suggest a USMB-induced increase in perfusion, but a Wilcoxon matched-pairs signed-rank test showed that medians before (17%) and after (19%) USMB therapy did not change significantly (*p* = 0.1099).

## 4. Discussion

We conducted a clinical feasibility study in a small cohort of feline companion animals, evaluating the combination of bleomycin chemotherapy and USMB therapy in cats with oral squamous cell carcinoma.

The USMB treatment sessions were feasible and well tolerated. Due to our patient selection process, all tumours were accessible to USMB therapy. Adverse events were considered to be related to anaesthesia, comorbidity or progressive tumour growth. The fibrosis-like functional impairment of the tongue leading to dysphagia grade 3 and finally euthanasia in cat 6 could be related to USMB and/or chemotherapy, but also to scar tissue of previous surgery. We did not observe severe adverse events, such as tumour lysis syndrome and disseminated intravascular coagulation, which have been described for electroporation [40]. Muscle contractions and arrhythmias are not expected to occur with USMB due to the difference in technique (ultrasound versus electrical pulses) and while vascular disruption caused by electroporation leads to acute tumour necrosis, USMB therapy may have a more gradual and tolerable anti-tumour effect. The relatively mild adverse effects of USMB therapy could imply that it is a more tolerable and feasible option than electroporation and that there is room for treatment intensification (i.e., more treatment sessions, higher dosages of chemotherapy or concurrent use of more than one chemotherapeutic agent).

Unfortunately, all cats except one eventually had progressive disease based on clinical tumour measurements. However, while the median overall survival is (historically) around 44 days with supportive care alone [33], survival in our study (albeit small and without a control group) was somewhat longer (46–85 days and 147 days in cat 6 where the tumour had been incompletely surgically removed before study treatment). Adding a control group treated with bleomycin alone could have provided more robust data on efficacy, but this was not feasible in our veterinary study. It can be expected that pet owners would not want to enroll their cat into a study when there was a chance of receiving the bleomycin alone, which is not a standard-of-care option and is reported to have limited efficacy in cats with this specific type of cancer. A previous study found a response rate of only 33% for bleomycin alone [41]. The possibility of a more effective treatment by addition of USMB led patient owners to choose study participation over supportive care alone. Since the primary goal of this study was to evaluate feasibility, the lack of a control group was considered acceptable. A number of hypotheses may explain why tumour responses and overall survival did not improve more substantially. First, the cavitation-induced permeabilization of cancer cells to enhance the intracellular uptake of bleomycin, which was previously demonstrated in vitro [50,51,52,53], may not have worked as well in vivo in our veterinary study. Unfortunately, we could not measure sonopermeation efficacy directly, i.e., by quantifying bleomycin uptake in tumours, nor indirectly, by monitoring microbubble cavitation activity during the procedure. Future studies using simultaneous USMB therapy and cavitation detection could confirm our assumption and lead to further optimization of US settings. Based on previous literature, this is feasible using a clinical US system, but it requires modifying predefined factory settings, which could lengthen the process towards clinical approval [58]. Second, USMB therapy will only cause sonopermeation in cells in the close proximity of microbubbles [59], i.e., endothelial cells and perhaps a few layers of adjacent tumour cells. Moreover, CEUS imaging showed that microbubbles did not spread throughout the entire tumour, which further diminishes the efficacy of sonopermeation. Third, the US parameters we used (which are standard settings for PW Doppler on the clinical US system) are likely to be suboptimal. Our previous in vitro results demonstrated that PW Doppler using an unmodified clinical ultrasound probe with the lowest centre frequency (S5-1 with a frequency of 1.6 MHz) and the maximum number of cycles per pulse (46) resulted in the most efficient cell permeabilization [53]. While it was not feasible to use the exact same settings in this veterinary study (the S5-1 probe provided insufficient anatomical detail for target identification in the cat), we were able to closely mimic them. Optimization steps were performed during the first treatment sessions. In the first treatment session of cat 1, a lower MB dosage and the S5-1 probe were used. In all treatment sessions of cat 1 and the first treatment session of cat 2, a higher mechanical index was applied (relative intensity 0 dB, MI = 1.0–1.2). However, decreased perfusion was noted on CEUS after USMB therapy in the first treatment session of cat 2; therefore, the relative intensity was decreased to −10 dB (i.e., MI = 0.3–0.4). With this drop of MI, we expect that we moved from an inertial cavitation to a stable cavitation regime. On the other hand, customized settings, such as those used by Keller et al. in a healthy porcine model, may provide even better results [43]. Fourth, if sonopermeation did occur successfully, (part of) these tumours may not have been intrinsically sensitive to bleomycin, meaning that bleomycin could not kill the cell even after entering it [60]. Finally, it is possible that more bleomycin plus USMB treatment sessions are needed to obtain a durable clinical response. It would be interesting to continue study treatment until tumour progression or unacceptable toxicity in a future study.

Although clinical response in our small study was modest, our findings of possibly increased tumour perfusion after USMB treatment were promising. While the median AUC increased and decreased in an equal number of sessions, the increase in median AUC was much larger. Increased AUC is a typical feature of increased perfusion, which indicates increased microvascular blood volume (MBV: the proportion of tissue volume existing of blood) [61,62]. Increased perfusion should also lead to shorter TTP, but this was not observed in our study. Improving MBV will benefit the exchange of oxygen, nutrients and drugs [63]. Note that these findings have to be interpreted with caution, as the number of evaluable treatment sessions is small and the position of the transducer (i.e., anatomical location) for CEUS ROIs before and after USMB was matched to the best of our ability but was never identical. Similar transducer positions for CEUS in different treatment sessions of the same cat are even harder to obtain. It is also possible that in cat 6 (in whom tumour perfusion apparently decreased in two out of three sessions), the effect of USMB was different because cat 6 was the only cat who had received surgery before study treatment. Nevertheless, previous studies have also shown that USMB therapy can affect tissue perfusion and thereby reduce tumour hypoxia. Both increased and decreased perfusion have been described, and more research is needed to determine which ultrasound parameters induce either effect. Decreased perfusion seems to be related to vascular damage and platelet activation, while increased perfusion is associated with vessel dilation and (in the longer term) induction of angiogenesis [64]. Improving tumour perfusion is also of interest for (chemo)radiotherapy in HNSCC since clinical response to chemoradiotherapy is negatively affected by tumour hypoxia [65], caused by structural abnormalities in the tumour vasculature [66,67]. Consequently, increased tumour perfusion could decrease hypoxia and thereby improve outcomes. Because the effect of bleomycin is mainly limited by intracellular uptake, rather than by perfusion, only a small survival benefit can be expected in our study due to increased tumour perfusion. In contrast, combining USMB therapy with drugs known to be perfusion-limited could lead to improved outcome in future studies.

Given the similarities between humans and cats in size and tumour characteristics, a feline veterinary trial provides a good opportunity to study the feasibility of USMB therapy using an unmodified clinical US system and clinically available microbubbles. Our optimized USMB therapy procedure can be easily translated to human patients. In the future, we expect that USMB therapy, if proven safe and feasible in human patients with head and neck cancer, could be added to standard-of-care chemo(radio)therapy to improve local drug delivery (e.g., of cisplatin and/or cetuximab) or to standard-of-care radiotherapy (to enhance tumour perfusion and consequently reduce hypoxia). Finally, adding USMB therapy to chemoradiotherapy could, in the future, lead to adapted treatment regimens with similar efficacy while reducing systemic toxicity. Early clinical trials using USMB in combination with chemotherapy in patients with various tumour types [20,21,22,23] (ClinicalTrials.gov Identifiers: NCT04146441, NCT04821284, NCT03477019 and NCT03458975, NCT03385200), as well as clinical studies of USMB therapy with radiotherapy in the absence of a drug (Clinicaltrials.gov NCT04431674, NCT04431648), have already shown promising results or are still ongoing. Since we used a clinical US system and EMA/FDA-approved microbubbles in this study, the step to a clinical trial in human head and neck cancer patients can be taken in the near future. While customized USMB therapy settings might lead to even better results in the long term, the approval process could take longer and we expect that the use of PW Doppler without adaptations will accelerate the road to clinical benefit.

## 5. Conclusions

Our veterinary feasibility trial shows that the combination of bleomycin and ultrasound and microbubbles therapy, using an unmodified clinical ultrasound system and FDA/EMA-approved microbubbles, is a feasible and well-tolerated treatment in cats with oral squamous cell carcinoma. Besides a modest clinical response, we found indications of enhanced tumour perfusion after USMB therapy. This could be a forward step toward clinical translation of USMB therapy to human patients with head and neck cancer or other tumours with a clinical need for locally enhanced treatment.

## Figures and Tables

**Figure 1 pharmaceutics-15-01166-f001:**
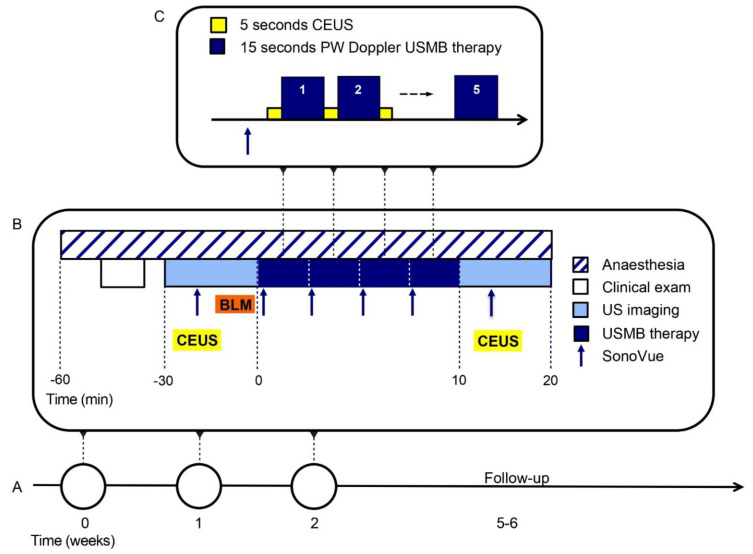
Ultrasound and microbubbles (USMB) treatment procedures. (**A**) Each cat is treated three times, once per week. (**B**) Each treatment session is performed under general anaesthesia and starts with a clinical exam. Ultrasound imaging and contrast-enhanced ultrasound imaging (CEUS) are performed before and after USMB therapy. Seven minutes after intravenous injection (i.v.) of bleomycin (BLM), USMB therapy is started by i.v. bolus injection of SonoVue microbubbles. (**C**) When the microbubbles appear on the CEUS image, 15 s of Pulsed Wave (PW) Doppler are alternated by 5 s of CEUS imaging. This is repeated five times per microbubble injection, for a total of four microbubble injections.

**Figure 2 pharmaceutics-15-01166-f002:**
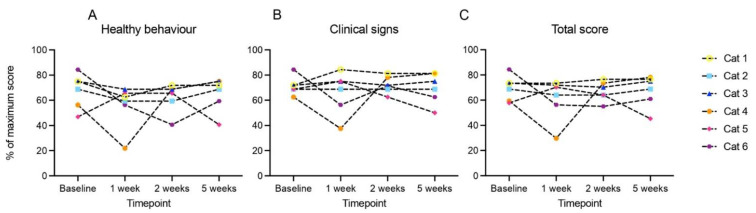
Quality of life per cat during the study, assessed by a 16-item owner-completed measure of feline quality of life, translated to Dutch with permission from Adelphi UK and Zoetis. Baseline was registered on the day of first treatment session, before treatment. Higher scores indicate better quality of life. (**A**) Healthy behaviour score, (**B**) clinical signs score, (**C**) total score, which is the mean of healthy behaviour and clinical signs.

**Figure 3 pharmaceutics-15-01166-f003:**
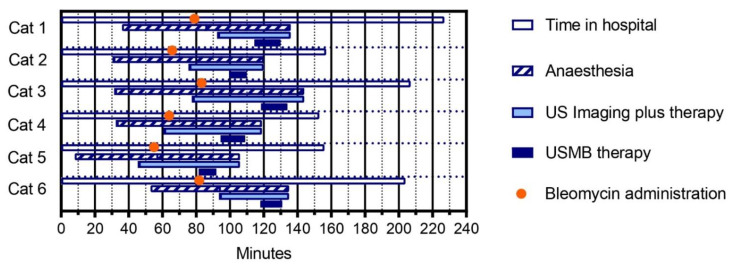
Feasibility of USMB treatment, assessed by duration of study procedures. For each procedure, the mean of three treatment days was calculated. In cat 1, treatment procedures had not yet been optimized.

**Figure 4 pharmaceutics-15-01166-f004:**
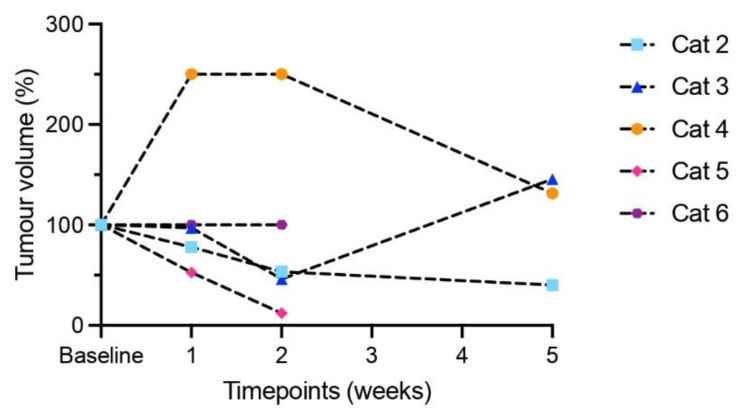
Estimation of tumour volumes (length × width × depth, percentage of baseline) in cats treated with optimized USMB settings, based on calliper measurements of the tumour prior to that day’s study treatment. Cats 2, 3 and 5 had stable disease during the three treatment sessions but showed disease progression 5 weeks (cats 3 and 5) or 11 weeks (cat 2, not shown here) after first treatment. In cat 5, reliable tumour measurement was not possible five weeks after treatment due to tumour necrosis, but clear disease progression was noted. Cat 4 had progressive disease one week after the first treatment (possibly due to tumour infection) but remained stable at follow-up. Cat 6 previously had macroscopical resection of the tumour prior to study inclusion, which made it difficult to measure the remaining tumour. Two-dimensional measurements showed stable disease until the moment of submission of this article and the cat is still alive 70 days after first USMB treatment.

**Figure 5 pharmaceutics-15-01166-f005:**
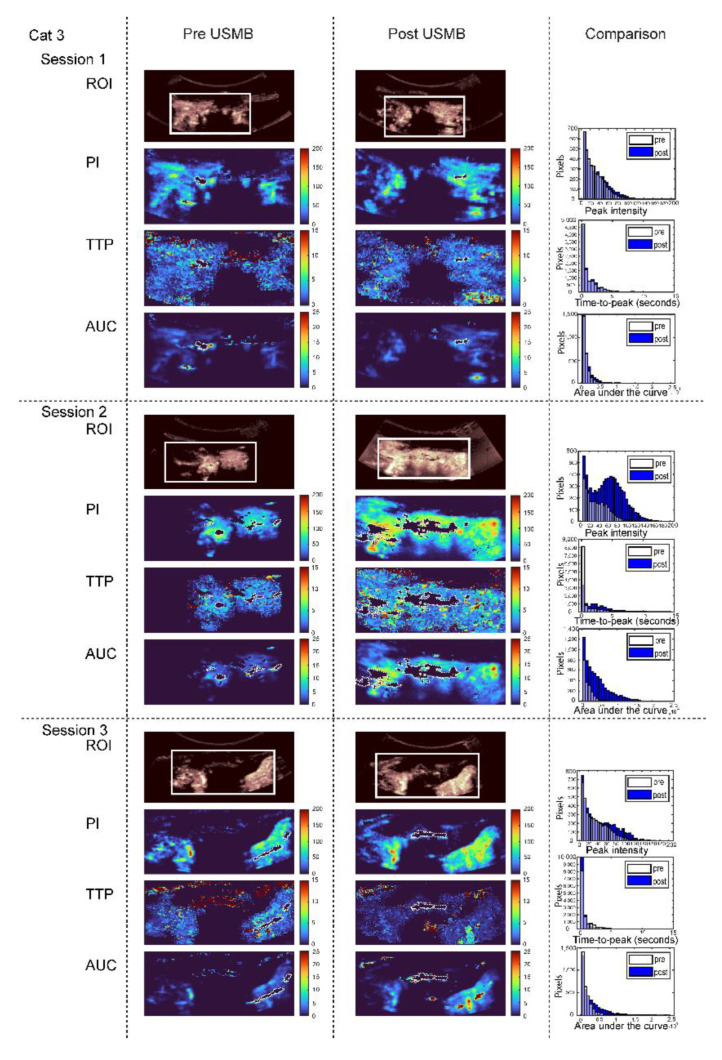
Contrast-enhanced ultrasound (CEUS) parametric maps of cat 3 before (first column) and after (second column) USMB therapy for visual comparison. For each of the three treatment sessions, from top to bottom, the following parameters are depicted: position of region of interest (ROI), peak intensity (PI), time to peak (TTP), area under the curve (AUC). Pixels with high signal intensity before administration of microbubbles (e.g., regions containing bone) were set to zero and are delineated with white dotted lines. In the third column these parameters are compared in histograms before (white) and after (blue) USMB. Note that the range of the y-axes differs between different treatment sessions. ROIs pre and post USMB are identical in size within one treatment session and matched in position as much as possible. CEUS parameters were kept constant between treatment sessions, except in treatment session 2 of cat 3 when gain inadvertently changed from 45% (before USMB) to 49% (after USMB).

**Figure 6 pharmaceutics-15-01166-f006:**
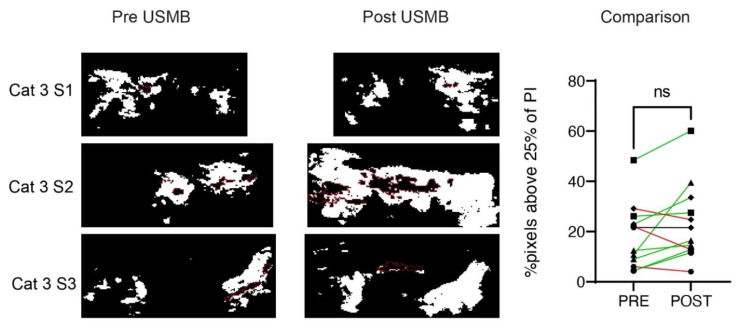
Peak Intensity (PI) on CEUS in the tumour before (**left**) and after (**right**) USMB therapy. Representative maps of cat 3 in three treatment sessions (S1–S3). Pixels with high signal intensity before administration of microbubbles (e.g., regions containing bone) were set to zero and are delineated with red dotted lines. The percentage of pixels reaching >25% of the peak intensity (white pixels) in the ROI was not significantly higher after USMB therapy compared to baseline. Green lines indicate an increase in the percentage of pixels reaching >25% of PI after USMB therapy compared to baseline in that treatment session, red lines indicate a decrease and black lines indicates no change. ns = Wilcoxon matched-pairs signed-rank test *p* = 0.1099.

**Table 1 pharmaceutics-15-01166-t001:** Optimized Pulsed Wave Doppler settings for USMB therapy and CEUS parameters used in cats 2–6.

Parameter	Indication/Setting on EPIQ5 or EPIQ7	Value
**Optimized Pulse Wave (PW) Doppler settings for ultrasound and microbubble (USMB) therapy**		
**Frequency**	C9-2 probe in PW mode	2.9 MHz
**Pulse length**	Sample volume: 7.5 mm (maximum)	21 cycles per pulse
**Pulse repetition frequency**	Scale: −4–4 cm/s (minimum)	0.4 kHz
**Mechanical index**	Relative intensity: −10 dB *	MI 0.3–0.4 at target depth
**Contrast-enhanced ultrasound (CEUS) settings**		
**Mechanical index (MI)**	MI in CEUS mode <0.1	CEUS MI = 0.06
**Gain**	Gain slightly above the noise floor in absence of microbubbles and kept constant	Gain = 45%
**Dynamic range (compression)**		Dynamic range = 50
**Focus position**	Focus positioned at the target or a bit deeper (2/3 of image depth)	Adjusted per treatment session and moved during USMB therapy
**Time gain compensation (TGC)**	All switches in central position	All switches in central position
**Persistence**		Off

* In cat 4 relative intensity was increased to −7 dB to account for enhanced attenuation due to extensive bone invasion and fibrous tissue formation of bone.

**Table 2 pharmaceutics-15-01166-t002:** Relevant patient and tumour characteristics, including survival.

Patient	1	2	3	4	5	6
Sex	male	male	male	female	male	female
Age at inclusion (years)	11	11	18	15	14	15
Body weight at inclusion (kg)	2.8	7.2	3.4	2.8	6.7	5.5
TNM stage [57]	T2N1M0	T2bN0M1 (lungs)	T2bN0M0	T3bN0Mx	cT3N1Mx	T1N0M0 *
Tumour location	tongue, frenulum, and sublingual soft tissue	right maxilla	lip and cheek extending into corner of mouth and caudal maxilla	rostrally in the mouth, infiltrated into mandibula	tongue base and floor of mouth	Sublingual, floor of mouth
Supportive care measures	Antibiotics Analgesics (NSAIDs, tramadol) Tube feeding	Antibiotics Analgesics (NSAIDs)	Antibiotics Analgesics (NSAIDs, gabapentin)	Antibiotics Analgesics (NSAIDs, gabapentin)	Antibiotics Analgesics (NSAIDs, buprenorphine) Tube feeding	Antibiotics Analgesics (NSAIDs) Antiemetics (maropitant) mirtazapine
Concomitant drugs	Treatment for hyperthyroidism (carbimazole) initiated during study	Treatment for hyperthyroidism (carbimazole) initiated during study	-	-	-	Treatment for hyperthyroidism (thiamazole) initiated before start of study
Survival (days)	46	85	64	56	57	147
Death	euthanasia	euthanasia	euthanasia	euthanasia	natural death	euthanasia

* Cat 6 had surgery before inclusion, the sublingual tumour was macroscopically removed and no longer visible. At inclusion, the floor of mouth felt firm over an area of <2 cm diameter (therefore the tumour was classified as T1).

**Table 3 pharmaceutics-15-01166-t003:** Adverse events.

Adverse Event (VCOG CTCAE Version 1.1)	Grade 1	Grade 2	Grade 3	Unknown Grade	Most Likely Related to
alopecia	1				i.v. catheter
anorexia		1	1		tumour progression
cardiac murmur				1	comorbidity (hyperthyroidism)
constipation	1				anaesthesia
dysphagia (fibrosis-like tissue)			1		possible reaction to USMB, chemotherapy or previous surgery
generalized muscle weakness	1				tumour progression
haemorrhage/bleeding	1				tumour progression
hypotension	1				anaesthesia
hypothermia		1			anaesthesia
lethargy/fatigue/decreased general performance	2	3			anaesthesia or tumour progression
localized erythema	1				i.v. catheter
oral ulcers	1				infection
pain (most likely at tumour site)		2			tumour progression
ptyalism		1			tumour progression
sinus tachycardia	1				possible reaction to USMB or anaesthesia
skin ulceration	1				tumour progression
soft tissue necrosis	1	2			tumour progression in cat 1 and 5, possible reaction to USMB or chemotherapy in cat 6
vomiting	1				anaesthesia
weight loss	2	1			tumour progression
**Any adverse event**	**15**	**11**	**2**	**1**	

## Data Availability

The data presented in this study are available on request from the corresponding author. The data are not publicly available due to privacy-sensitivity.

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
