# Peer review of "Ultrasound and Microbubbles Mediated Bleomycin Delivery in Feline Oral Squamous Cell Carcinoma—An In Vivo Veterinary Study"

_pharmaceutics, 2023, doi:10.3390/pharmaceutics15041166_

Round 1
Reviewer 1 Report
The Manuscript entitled “ultrasound and microbubbles mediated bleomycin delivery in feline oral squamous cell carcinoma – a veterinary study” by Maar et al. is interesting. This manuscript is good in quality, novelty, and writing so definitely suitable for publication in Pharmaceutics. But the minor comments need to be addressed by the author before publishing it.
Comment to author
Why does the author mention a veterinary study, why not it is an in vivo study? Just need to clarify the difference.
What are the death statistics due to head and neck cell carcinoma?
In Figure 2, what represents the baseline? Need to clarify.
If the author conducts a triplicate study, they need a standard deviation in Figures 2 and 4.
Figures 2 and 4 could be more attractive when they are in color. Need to take care.
The author needs to be consistent in the figures and the font used in the figures. Compared to the figure 2 font, the figure 4 font is too big, and some other figures have extremely low font sizes.
The author's contribution is missing after the conclusion section.
Funding details also need to be disclosed.
As this study is related to veterinary, if any project number needs to acknowledge as well as any permission required to conduct such study, approvals need to be disclosed If any
Reviewer 2 Report
In this paper, the authors have first shown the safety and tolerability of microbubble-assisted bleomycin delivery to head and neck squamous carcinoma (HNSC) tumors. The authors have presented the results of HNSC tumor response to sonoporation combined with systemic bleomycin injection. The authors report a slight survival increase compared to standard care and conclude that microbubble sonoporation drug delivery is feasible and tolerable in cats with HNSC tumors which is a promising result and possibly translatable to clinics.
I believe that this area of research has merit since the literature in ultrasound drug delivery field is mainly focused on brain delivery, and papers like this that primarily focus on other common cancer types are needed. The authors did an excellent job with the introduction and perfectly set the problem that led to this research. However, I have a few comments that I believe need to be addressed before publishing this work.
1. The major area of concern is the fact that this research has a single arm. In order to get a valid conclusion, a negative control is needed. Unfortunately, not having negative control casts doubt on the paper's overall conclusion. A perfect negative control will be a group of cats that only receive bleomycin without ultrasound. Adding more negative control groups can even make a stronger case for the conclusions derived. The authors need to explain their rationale for having a single arm for this experiment at a minimum.
2. Although I believe the paper has a strong introduction, I think it is better to include other papers that have used bleomycin with sonoporation previously (if any) in the intro.
3. Figure 5 on the paper is a comprehensive one; however, its quality is not optimal. Increasing the font size on the comparison column might be helpful
4. I think it will be helpful if the authors add a paragraph on bleomycin's chemical and physical properties (such as hydrophobicity or pharmacokinetics) in the intro or discussion.
Round 2
Reviewer 2 Report
I have no more comments